# A cross-modality enhancement of defensive flight via parvalbumin neurons in zona incerta

Xiyue Wang[1,2†], Xiaolin Chou[1,2†], Bo Peng[1,2], Li Shen[1], Junxiang J Huang[1,3], Li I Zhang[1,4]*, Huizhong W Tao[1,4]*

[1]Zilkha Neurogenetic Institute, Keck School of Medicine, University of Southern California, Los Angeles, United States; [2]Graduate Program in Neuroscience, University of Southern California, Los Angeles, United States; [3]Graduate Program in Biomedical and Biological Sciences, University of Southern California, Los Angeles, United States; [4]Department of Physiology and Neuroscience, Keck School of Medicine, University of Southern California, Los Angeles, United States

**Abstract** The ability to adjust defensive behavior is critical for animal survival in dynamic environments. However, neural circuits underlying the modulation of innate defensive behavior remain not well-understood. In particular, environmental threats are commonly associated with cues of multiple sensory modalities. It remains to be investigated how these modalities interact to shape defensive behavior. In this study, we report that auditory-induced defensive flight behavior can be facilitated by somatosensory input in mice. This cross-modality modulation of defensive behavior is mediated by the projection from the primary somatosensory cortex (SSp) to the ventral sector of zona incerta (ZIv). Parvalbumin (PV)-positive neurons in ZIv, receiving direct input from SSp, mediate the enhancement of the flight behavior via their projections to the medial posterior complex of thalamus (POm). Thus, defensive flight can be enhanced in a somatosensory context-dependent manner via recruiting PV neurons in ZIv, which may be important for increasing survival of prey animals.
DOI: https://doi.org/10.7554/eLife.42728.001

*For correspondence:
liizhang@usc.edu (LIZ);
htao@usc.edu (HWT)

†These authors contributed equally to this work

Competing interests: The authors declare that no competing interests exist.

## Introduction

Defensive behaviors are critical for animal survival. They are dynamic and adaptive, as environmental contexts, properties and intensity of threats, as well as expectations from past experiences can all modulate the form as well as the magnitude of defensive behaviors (*Fanselow, 1994*; *Gross and Canteras, 2012*; *Tovote et al., 2016*). Threat signals in the external environment are sensed by different sensory modalities through distinct sensory pathways to initiate appropriate defensive behaviors. Previous studies have mostly been focused on defensive behaviors initiated under stimulation of one individual sensory modality (*Fanselow and LeDoux, 1999*; *Yilmaz and Meister, 2013*). However, a danger may be associated with cues of multiple sensory modalities arriving at the same time, and the integration of information of these different modalities may profoundly influence the behavioral output. Intuitively, the presence of multisensory signals is helpful for strengthening defensive responses. However, neural circuit bases for the potential cross-modality interactions in defensive behaviors are largely unknown. In this study, we designed experiments to specifically examine whether tactile input can affect a well-established auditory-induced defensive behavior (*Fanselow and LeDoux, 1999*; *Tovote et al., 2016*). The vibrissal system is crucial to behaviors such as navigation and exploration (*Carvell and Simons, 1990*; *Diamond et al., 2008*), and rodents frequently collect information from surroundings using their whiskers (*Prigg et al.,*

*2002*). We reason that it may be common for animals to use both vibrissal and auditory systems in sensing environmental dangers.

Zona incerta (ZI) is a major GABAergic subthalamic nucleus consisting of heterogeneous groups of cells. In rodents, four (rostral, ventral, dorsal, caudal) sectors of ZI can be loosely defined based on neurochemical expression patterns (*Ma et al., 1997*; *Mitrofanis et al., 2004*), and it has been suggested that different sectors might be involved in different circuits and functions (*Liu et al., 2017*; *Plaha et al., 2008*). Our recent study has shown that GABAergic neurons in the rostral sector of ZI (ZIr) play a role in reducing defensive behavior in an experience-dependent manner (*Chou et al., 2018*). It also raises a possibility that ZI might play a broader role in defensive behavior. ZI receives inputs from various cortical areas including the primary somatosensory cortex (SSp) (*Kolmac et al., 1998*; *Shammah-Lagnado et al., 1985*) as well as from the brainstem trigeminal nucleus that relays vibrissal information (*Roger and Cadusseau, 1985*; *Smith, 1973*). A recent study has demonstrated that deflecting whiskers directly induces neuronal activity in the ventral sector of ZI (ZIv) (*Urbain and Deschênes, 2007*), where parvalbumin (PV) positive neurons are enriched (*Kolmac and Mitrofanis, 1999*). In the present study, we investigated whether somatosensory input through whisker stimulation could modulate defensive behavior via recruiting ZIv PV + neurons.

## Results

To test whether tactile input can affect defensive behavior, we employed a relatively simple behavioral test, sound-induced flight, following our previous studies (*Xiong et al., 2015*; *Zingg et al., 2017*). Such behavior has been observed in both freely moving and head-fixed conditions (*Xiong et al., 2015*; *Zingg et al., 2017*). In our first set of experiments, animals were head-fixed and placed on a smoothly rotatable plate (*Chou et al., 2018*; *Liang et al., 2015*). Loud noise sound (80 dB sound pressure level or SPL) elicited animal running, and the running speed was recorded in real time (*Figure 1A*, left). Tactile stimulation was applied by deflecting whiskers unilaterally with a cotton stick controlled by a piezo actuator (*Figure 1—figure supplement 1A*). In our control experiments, the whisker deflection per se did not elicit significant locomotion of animals (*Figure 1—figure supplement 1B*). Trials without and with tactile stimulation were interleaved. We found that tactile stimulation enhanced the running induced by noise sound (*Figure 1A*, right), as demonstrated by the increased peak speed (*Figure 1B*, *Figure 1—figure supplement 1C*) and total travel distance (*Figure 1C*, *Figure 1—figure supplement 1D*). The temporal profile of the behavioral response was not significantly affected, as shown by the quantifications of onset latency and time to peak (*Supplementary file 1*). Silencing the SSp contralateral to the whiskers being deflected by infusing a GABA receptor agonist, muscimol (*Figure 1D*, left), removed the difference in speed between conditions without and with whisker stimulation (*Figure 1D–F*), without altering the response temporal profile (*Supplementary file 1*). This suggests that the tactile enhancement of running is mediated mainly through SSp. To further demonstrate the tactile effect on flight behavior in freely moving animals, we used a two-chamber test following our previous study (*Zingg et al., 2017*). When the mouse was exposed to noise applied in one chamber, it quickly escaped to the other chamber by crossing through a narrow channel (*Figure 1—figure supplement 2A*). Trimming of all whiskers of the animal significantly decreased the average speed of the flight through the channel (*Figure 1—figure supplement 2B*), suggesting that tactile information through whiskers can indeed enhance flight behavior in a more natural condition.

Previous studies have suggested that SSp projects to ZIv (*Kolmac et al., 1998*; *Shammah-Lagnado et al., 1985*), and that ZIv neurons respond to whisker deflections (*Urbain and Deschênes, 2007*). To confirm this projection, we injected AAV1-CamKII-hChR2-eYFP into SSp of PV-*ires*-Cre crossed with Ai14 (Cre-dependent tdTomato) reporter mice (*Figure 2A*). We found profuse fluorescence-labeled axons in ZIv, but few in other ZI sectors (*Figure 2B*). We next directly examined the effect of stimulating the SSp projection to ZIv, by placing optic fibers on top of ZIv to deliver LED light pulses (20 Hz train for 5 s) bilaterally (*Figure 2C*). The optogenetic activation of the SSp axons in ZIv enhanced noise-induced running (*Figure 2C–E*, *Figure 2—figure supplement 1A–B*) without affecting the response temporal profile (*Supplementary file 1*), but by itself had no effect on the baseline locomotion speed (*Figure 2—figure supplement 1C*). Infusing muscimol into ZIv bilaterally abolished the enhancement of flight response by whisker stimulation (*Figure 2F–H*) without affecting

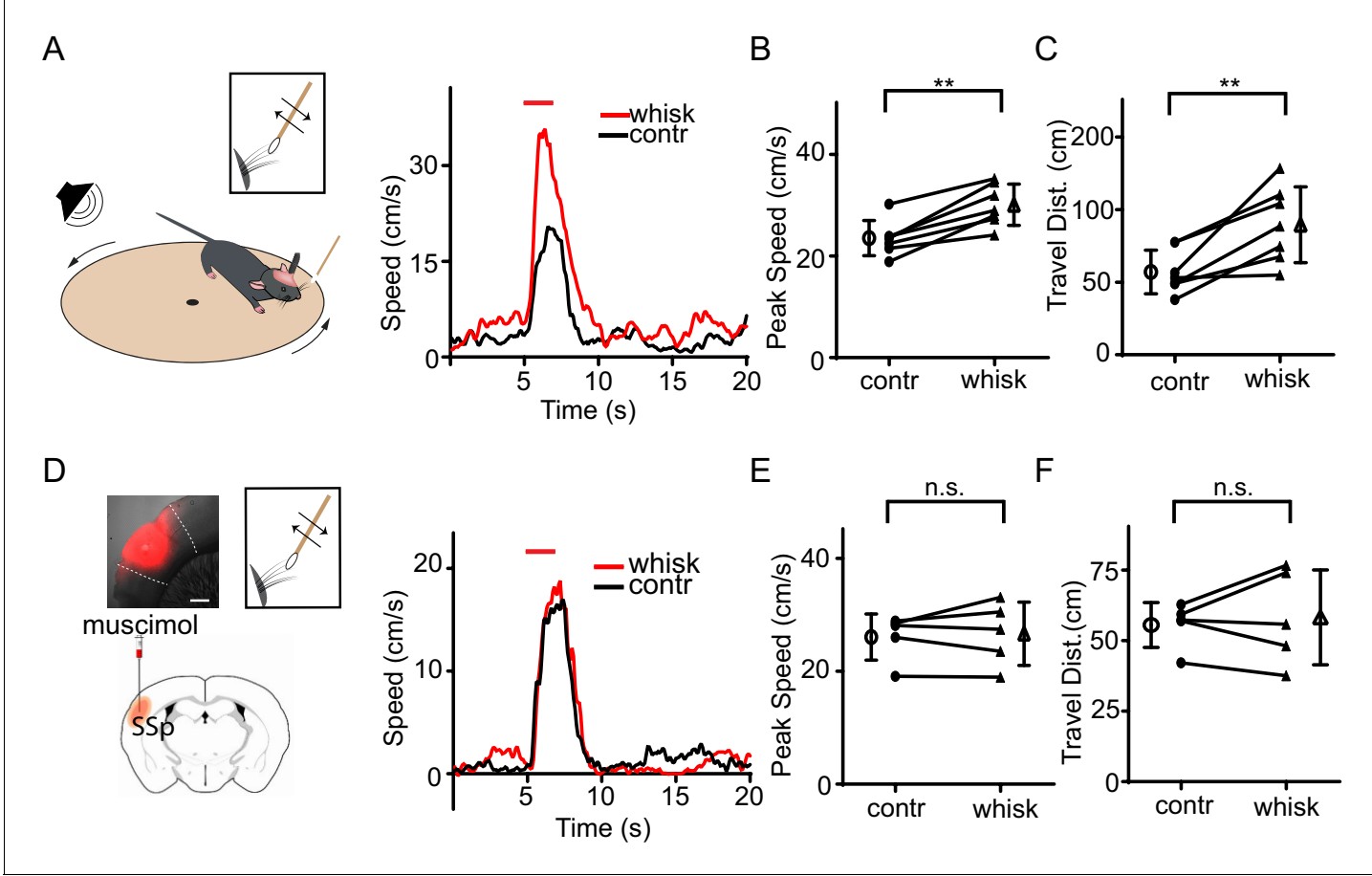

**Figure 1.** Tactile stimulation enhances sound-induced flight response via SSp. (A) Left, illustration of the head-fixed animal behavioral paradigm. Right, plots of running speed under noise presentation without (black) and with (red) concurrent whisker stimulation for an example animal. Red line marks the duration of noise/whisker stimulation. (B) Summary of peak noise-induced running speed in the absence and presence of whisker stimulation. **p=0.0011, two-sided paired t-test, n = 7 animals. (C) Summary of total travel distance. **p=0.0072, two-sided paired t-test, n = 7 animals. (D) Left, illustration of the experimental paradigm: SSp was silenced with infusion of muscimol (red) as shown in the confocal image (upper left, scale: 500 μm). Right, plots of speed without (black) and with (red) whisker stimulation for an example animal. (E) Summary of peak speed in the absence and presence of whisker stimulation. 'n.s.', not significant, two-sided paired t-test, n = 5 animals. (F) Summary of total travel distance. 'n.s.', not significant, two-sided paired t-test, n = 5 animals. Open symbols represent mean ± s.d. for all panels.

DOI: https://doi.org/10.7554/eLife.42728.002

The following source data and figure supplements are available for figure 1:

**Source data 1.** Data for *Figure 1* and *Figure 1—figure supplements 1* and *2*.
DOI: https://doi.org/10.7554/eLife.42728.005
**Figure supplement 1.** Control experiments and analysis of individual animals.
DOI: https://doi.org/10.7554/eLife.42728.003
**Figure supplement 2.** A flght test in freely moving mice.
DOI: https://doi.org/10.7554/eLife.42728.004

the response temporal profile (*Supplementary file 1*). Together, these results suggest that activation of the SSp-ZIv projection is sufficient and necessary for the tactile enhancement of auditory-induced flight response.

Immuno-histological studies have suggested that PV+ neurons are a major cell type in the ventral sector of ZI (*Kolmac and Mitrofanis, 1999*). To test whether SSp axons innervate PV+ neurons, we performed slice whole-cell recording from ZIv PV+ neurons (labeled by tdTomato expression in PV-Cre::Ai14 animals) while optically activating ChR2-expressing SSp axons in ZI (*Figure 2I*). We observed that blue light pulses evoked monosynaptic excitatory postsynaptic currents (EPSCs) in most ZIv PV+ neurons recorded with tetrodotoxin (TTX) and 4-aminopyridine (4-AP) present in the

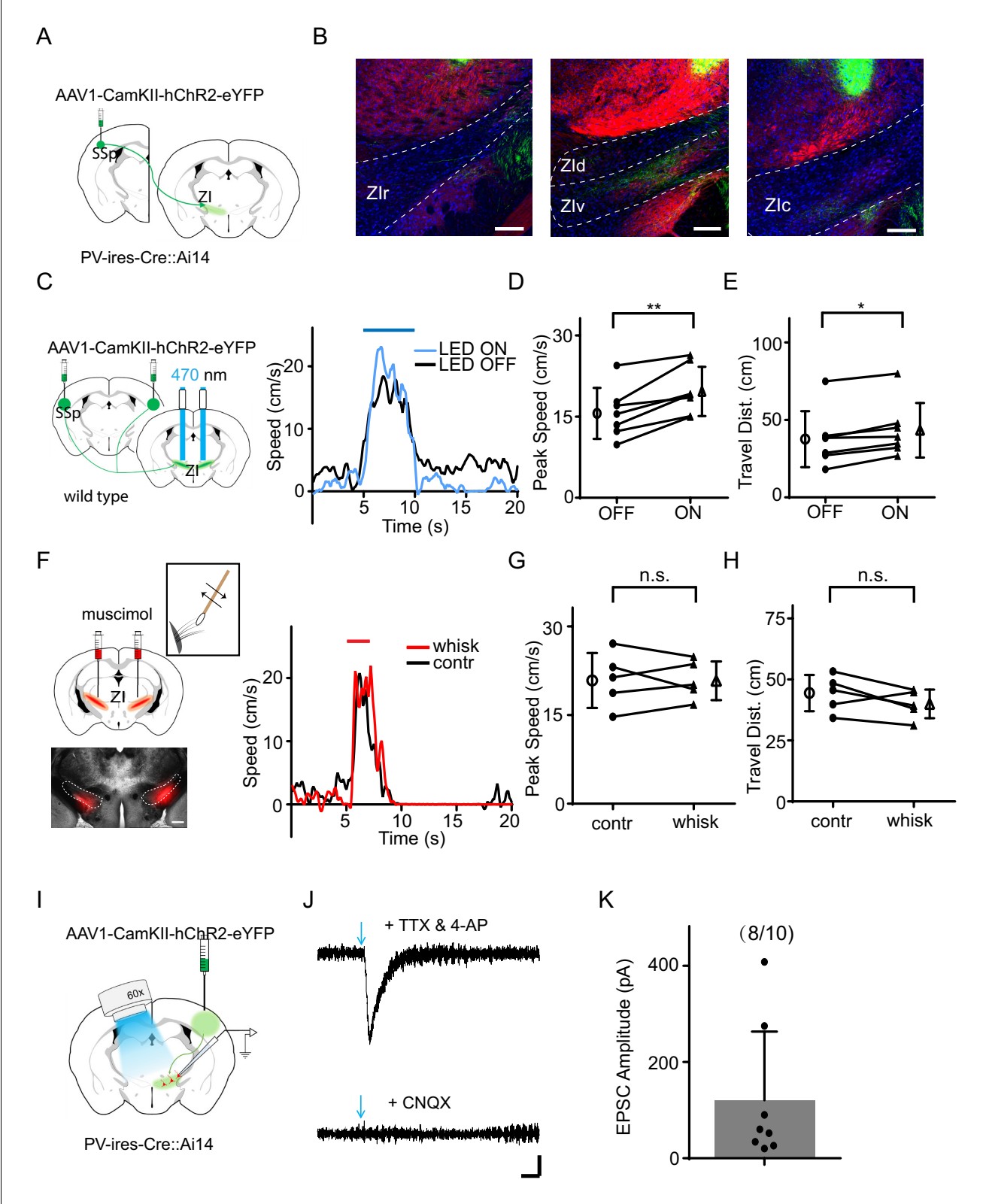

**Figure 2.** The SSp-ZIv projection mediates the tactile enhancement of sound-induced flight. (**A**) Illustration of the injection paradigm. (**B**) Anterogradely labeled axon terminals in rostral (left), dorsal and ventral (middle), as well as caudal (right) sectors of ZI. Scale bar, 200 μm. Blue shows Nissl staining; red shows PV+ neuron or axon distribution. (**C**) Left, illustration of the experimental paradigm: optic fibers were implanted to stimulate ChR2-expressing SSp axons in ZI. Right, plots of speed without (black) and with (blue) LED stimulation for an example animal. (**D**) Summary of peak

*Figure 2 continued on next page*

*Figure 2 continued*

noise-induced speed in the absence and presence of LED stimulation of SSp-ZI terminals. **p=0.003195, two-sided paired t-test, n = 7 animals. (E) Summary of the travel distance. *p=0.01854, two-sided paired t-test, n = 7 animals. (F) Left, ZIv was silenced with muscimol (red) as shown in the confocal image (lower, scale: 500 µm). Right, plots of speed without (black) and with (red) whisker stimulation for an example animal. (G) Summary of peak speed in the absence and presence of whisker stimulation. 'n.s.", not significant, two-sided paired t-test, n = 5 animals. (H) Summary of total travel distance. 'n.s.", not significant, two-sided paired t-test, n = 5 animals. (I) Experimental paradigm for slice recording. (J) Average LED-evoked EPSC in an example ZIv PV+ neuron before and after (lower) perfusion of CNQX. Arrow points to the onset of LED light. Recording was made in the presence of TTX and 4-AP. Scale: 25 pA, 25 ms. (K) Amplitudes of LED-evoked EPSCs of 8 responding neurons out of 10 recorded ZIv PV+ cells. Bars represent s.d. for all panels.

DOI: https://doi.org/10.7554/eLife.42728.006

The following source data and figure supplement are available for figure 2:

**Source data 1.** Data for *Figure 2* and *Figure 2—figure supplement 1*.
DOI: https://doi.org/10.7554/eLife.42728.008
**Figure supplement 1.** Analysis of individual animals and control experiment of LED stimulation alone.
DOI: https://doi.org/10.7554/eLife.42728.007

bath solution. The EPSC could be blocked by an AMPA receptor blocker, 6-cyano-7-nitroquinoxa-line-2,3-dione (CNQX) (*Figure 2J–K*). These results indicate that ZIv PV+ neurons receive direct excitatory input from SSp.

To investigate whether the PV+ neurons play a role in the tactile modulation of flight response, we injected AAV encoding Cre-dependent ChR2 or ArchT into ZI of PV-Cre::Ai14 mice (*Figure 3A, D*). The viral expression of opsins co-localized well with Cre-dependent tdTomato expression (*Figure 3—figure supplement 1A*), indicating PV-specific expression of opsins. Optogenetic manipulation of ZI PV+ neuron activity with blue (for the ChR2 group to activate) or green (for the ArchT group to suppress) LED light was interleaved with control trials in which only sound was delivered. The efficacies of ChR2 and ArchT were confirmed by slice whole-cell recordings showing that blue LED light evoked robust spiking in ChR2-expressing neurons and green LED light induced a strong hyperpolarization of the membrane potential in ArchT-expressing cells (*Figure 3—figure supplement 1B–C*). We found that activation of ZI PV+ neurons enhanced flight response induced by noise stimulation (*Figure 3A–C*, *Figure 3—figure supplement 1D–E*), whereas suppression of these neurons reduced the flight response (*Figure 3D–F*, *Figure 3—figure supplement 1F–G*). None of the manipulations affected the temporal profile of the behavioral response (*Supplementary file 1*). As a control, neither activation nor suppression of ZIv PV+ neurons alone significantly affected the baseline locomotion (*Figure 3—figure supplement 2*). We next expressed Cre-dependent inhibitory designer receptors exclusively-activated by designer drugs (DREADDi) (*Zhu and Roth, 2014*), hM4D (Gi), in ZI of PV-Cre mice, and intraperitoneally injected the DREADDi agonist, clozapine-N-oxide (CNO) (*Figure 3G*). The efficacy of DREADDi inhibition was confirmed by slice recording showing that CNO increased the threshold for spiking and decreased the number of spikes evoked by current injections (*Figure 3—figure supplement 3*). The chemogenetic silencing of ZIv PV+ neurons prevented the enhancement of noise-induced flight response by whisker stimulation (*Figure 3G–I*) without affecting the response temporal profile (*Supplementary file 1*).

We next performed awake single-unit optrode recordings in ZI, following our previous study (*Zhang et al., 2018*). ZIv PV+ neurons were optogenetically identified by their time-locked spike responses to blue laser pulses (*Figure 3J*). We found that these neurons responded to both noise sound and whisker deflections and that concurrent whisker deflections increased the response level to noise (*Figure 3K–L*). This result indicates that ZIv PV+ neurons can integrate tactile and auditory inputs and that tactile input plays a faciliatory role, consistent with the behavioral results. Altogether, our results strongly suggest that the tactile enhancement of flight behavior is mediated primarily by ZIv PV+ neurons.

To identify which downstream target nucleus of ZIv PV+ neurons is involved in the behavioral modulation, we traced the projections from ZIv PV+ neurons by injecting AAV encoding Cre-dependent GFP in PV-Cre mice (*Figure 4A*). Consistent with previous results (*Barthó et al., 2002*; *Trageser and Keller, 2004*), we found that two targets, the medial posterior complex of thalamus (POm) and superior colliculus (SC), received the strongest projections from ZIv PV+ neurons (*Figure 4B*, *Figure 4—figure supplement 1*). We then specifically activated the ZIv PV+ projection

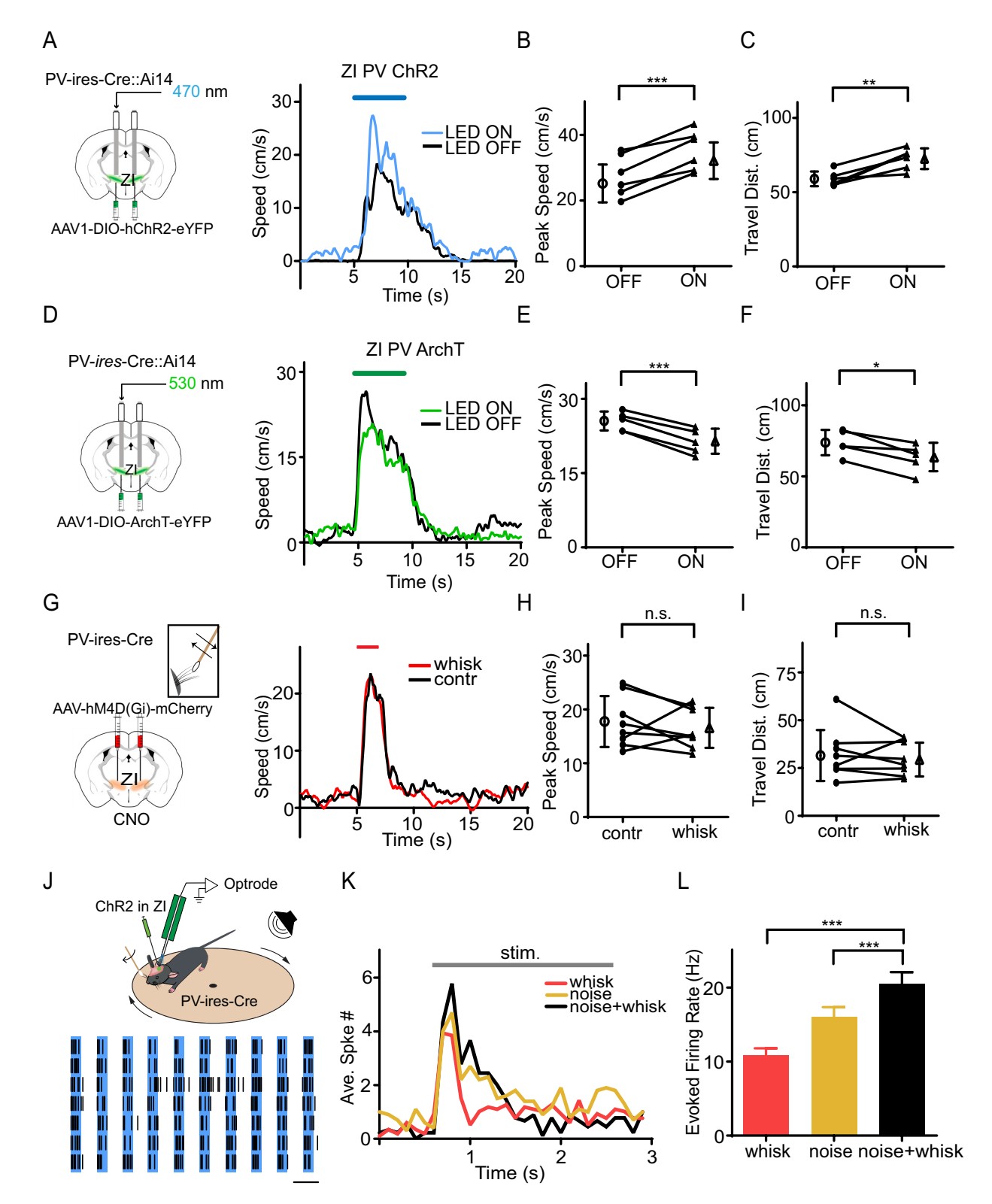

**Figure 3.** PV+ neurons in ZIv mediate the tactile enhancement of flight behavior. (**A**) Left, experimental paradigm. Right, Plots of speed without (black) and with (blue) LED stimulation for an example animal. Blue line marks the duration of noise/LED stimulation. (**B**) Summary of peak noise-induced speed in the absence and presence of LED stimulation of ZIv PV+ neurons. ***p=0.0009, two-sided paired t-test, n = 6 animals. (**C**) Summary of total travel distance. **p=0.0042, two-sided paired t-test, n = 6 animals. (**D**) Left, experimental paradigm. Right, plots of speed without (black) and with

*Figure 3 continued on next page*

*Figure 3 continued*

(green) LED stimulation for an example animal. Green line marks the duration of noise/LED stimulation. (E) Summary of peak noise-induced speed in the absence and presence of LED inhibition. ***p=0.0004, two-sided paired t-test, n = 5 animals. (F) Summary of total travel distance. *p=0.0136, two-sided paired t-test, n = 5 animals. (G) Left, expressing DREADDi in ZIv PV+ neurons. Right, plots of speed without (black) and with (red) whisker stimulation for an example animal. (H) Summary of peak noise-induced speed in the absence and presence of whisker stimulation with ZIv PV+ neurons inhibited by CNO. 'n.s.", not significant, two-sided paired t-test, n = 8 animals. (I) Summary of total travel distance. 'n.s.", not significant, two-sided paired t-test, n = 8 animals. Open symbols represent mean ± s.d. (J) Upper, optrode recording in the head-fixed animal. Lower, raster plot of an example ZIv PV+ neuron to 20 Hz LED stimulation in seven trials. Scale: 50 ms. (K) Peri-stimulus spike time histogram for an example PV+ neuron in response to whisker (red), noise (yellow) and whisker plus noise (black) stimulation. Bin size = 100 ms. (L) Summary of evoked firing rates of recorded PV + neurons (within the stimulation window). ***p<0.0001, one-way ANOVA with post hoc test, n = 22 cells.
DOI: https://doi.org/10.7554/eLife.42728.009

The following source data and figure supplements are available for figure 3:

**Source data 1.** Data for *Figure 3* and *Figure 3—figure supplements 1–3*.
DOI: https://doi.org/10.7554/eLife.42728.013
**Figure supplement 1.** Tests of efficacies of ChR2 and ArchT stimulation and analysis of individual animals.
DOI: https://doi.org/10.7554/eLife.42728.010
**Figure supplement 2.** Control experiments of LED stimulation alone.
DOI: https://doi.org/10.7554/eLife.42728.011
**Figure supplement 3.** Test of efficacy of chemogenetic silencing.
DOI: https://doi.org/10.7554/eLife.42728.012

to POm or SC by placing optic fibers on top of POm or SC, respectively, in PV-Cre mice injected with AAV encoding Cre-dependent ChR2 in ZI (*Figure 4C,F*). While activation of the ZIv-SC projection did not significantly change the flight speed (*Figure 4C–E*), that of the ZIv-POm projection significantly increased the flight speed (*Figure 4F–H*, *Figure 4—figure supplement 2A–B*), similar to the activation of ZIv PV+ neuron cell bodies. As a control, activation of the ZIv-POm projection alone did not change the baseline locomotion speed (*Figure 4—figure supplement 2C*). To confirm that the ZIv-POm projection is necessary for the tactile modulation, we expressed Cre-dependent hM4D (Gi) in ZI of PV-Cre mice and locally infused CNO into POm through implanted cannulas (*Figure 4I*). The chemogenetic silencing of the ZIv-POm projection prevented the enhancement of flight speed by whisker stimulation (*Figure 4I–K*). None of the manipulations affected the temporal profile of flight response (*Supplementary file 1*). Taken together, our results demonstrate that the projection of ZIv PV+ neurons to POm primarily mediates the enhancement of sound-induced flight behavior by tactile stimulation.

## Discussion

In this study, we demonstrate that additional tactile stimulation enhances flight behavior triggered by threats such as loud noise. Both SSp and ZIv PV+neurons, which receive SSp input, are necessary for this modulation, and activation of the SSp-ZIv projection is sufficient for driving the enhancement of the behavior. We also demonstrate that activation of ZIv PV+ neurons alone can enhance the flight behavior and that inactivation of the PV+ neurons or their projections to POm blocks the tactile enhancement of the flight behavior. Together, our data suggest that tactile input through whisker deflections can modulate defensive flight via the SSp-ZIv-POm pathway.

Rodents frequently use their whiskers to locate and identify objects (*O'Connor et al., 2013*; *Pammer et al., 2013*). In complex sensory environments, whiskers are essential for them to gather information from surroundings as to guide their behaviors during exploration and navigation (*Ahl, 1986*; *Diamond et al., 2008*; *Sofroniew et al., 2014*; *Yu et al., 2016*). When escape behavior is concerned, specific somatosensory input plus loud sound may indicate the immediate proximity of a predator, and enhancement of flight at such moments will greatly increase survival chances of prey animals. In addition, information conveyed by the somatosensory system about the environment could be extremely useful for the prey animal to quickly choose an effective escape route (*Diamond et al., 2008*; *Douglass et al., 2008*). Therefore, the ability to integrate somatosensory input and modulate defensive flight behavior accordingly is beneficial for animals to avoid dangers. Here, we show that somatosensory input from whiskers can enhance auditory-induced flight

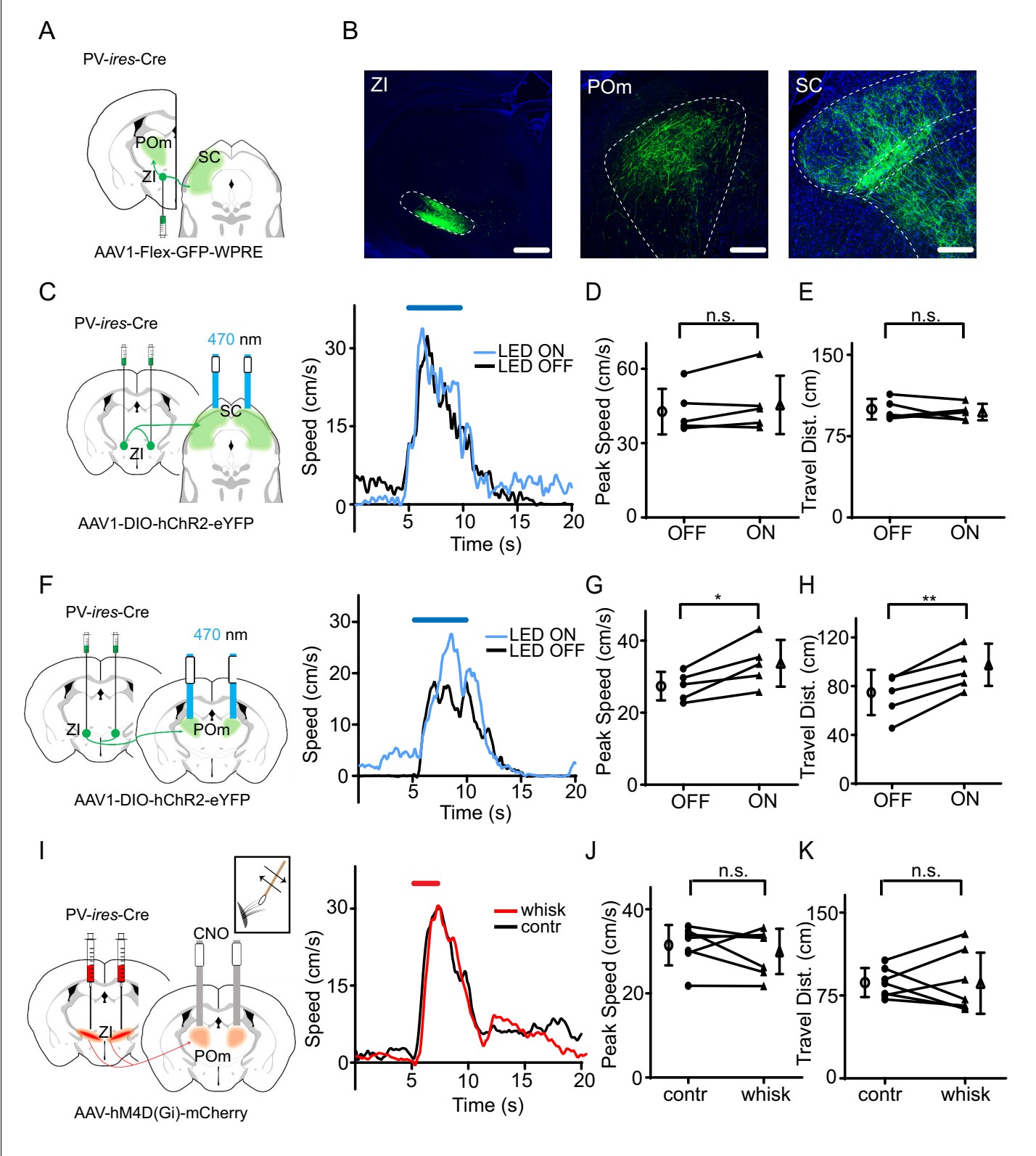

**Figure 4.** The projection of ZIv PV+ neurons to POm enhances sound-induced flight. (**A**) Illustration of injection paradigm. (**B**) Confocal images showing GFP expression at the injection site (left; scale: 500 µm) and in major target regions (middle and right; scale: 200 µm). Blue shows Nissl staining. SC, superior colliculus; POm, posterior medial nucleus of thalamus. (**C**) Left, stimulating ChR2-expressing ZI PV+ neuron axons in SC. Right, plots of speed without (black) and with (blue) LED stimulation for an example animal. (**D**) Summary of peak noise-induced speed in the absence and presence of LED

*Figure 4 continued on next page*

*Figure 4 continued*

activation of ZIv-SC axons. 'n.s.", not significant, two-sided paired t-test, n = 5 animals. (E) Summary of total travel distance. Two-sided paired t-test, n = 5 animals. (F) Left, stimulating ChR2-expressing ZI PV+ neuron axons in POm. Right, plots of speed without (black) and with (blue) LED stimulation for an example animal. (G) Summary of peak noise-induced speed in the absence and presence of LED activation of ZI-POm axons. *p=0.0198, two-sided paired t-test, n = 5 animals. (H) Summary of total travel distance. **p=0.0034, two-sided paired t-test, n = 5 animals. (I) Left, silencing DREADDi-expressing ZI PV+ neuron axons in POm. Right, plots of speed without (black) and with (red) whisker stimulation after local infusion of CNO in POm for an example animal. (J) Summary of peak noise-induced speed in the absence and presence of whisker stimulation when silencing ZIv-POm axons. 'n.s.', not significant, two-sided paired t-test, n = 7 animals. (K) Summary of total travel distance. 'n.s.', not significant, two-sided paired t-test, n = 7 animals. Open symbols represent mean ± s.d. for all panels.

DOI: https://doi.org/10.7554/eLife.42728.014

The following source data and figure supplements are available for figure 4:

**Source data 1.** Data for *Figure 4* and *Figure 4—figure supplements 1* and *2*.
DOI: https://doi.org/10.7554/eLife.42728.017
**Figure supplement 1.** Quantification of relative fluorescence density of GFP-labeled processes in different downstream regions of ZIv PV+ neurons (n = 4 animals).
DOI: https://doi.org/10.7554/eLife.42728.015
**Figure supplement 2.** Analysis of individual animals and control experiment of LED stimulation alone.
DOI: https://doi.org/10.7554/eLife.42728.016

response. Indeed, in freely moving mice, trimming of whiskers reduces the efficiency of their escape from a source of loud noise by crossing through a channel, indicating a faciliatory role of the tactile input.

The tactile-auditory cross-modality modulation relies on conveying somatosensory information primarily from SSp to ZI. ZI has been implicated in maintaining normal posture and locomotor functions (*Edwards and Isaacs, 1991*), as it sends dense projections to motor-related thalamic and brainstem nuclei (*Kolmac et al., 1998*; *Shaw and Mitrofanis, 2002*). The somatosensory input to ZI thus has a potential to influence motor functions (*Périer et al., 2002*; *Supko et al., 1991*). In this study, we show that SSp projections to ZI are mainly concentrated in ZIv, where PV+ neurons are a major cell type (*Mitrofanis, 2005*; *Zhou et al., 2018*). Consistent with this projection, PV+ neurons in ZIv receive direct excitatory input from SSp and respond to whisker deflections. Concurrent whisker deflections also increase their responses to sound, indicating that tactile-auditory integration takes place in ZIv PV+ neurons. Activating SSp-ZIv axon terminals or ZIv PV+ neurons directly enhances auditory-induced flight, while silencing the PV+ neurons abolishes the enhancement of flight by tactile stimulation. Therefore, our data demonstrate that via the SSp-ZIv pathway mediated mainly by ZIv PV+ neurons, somatosensory input can modulate motor functions in defensive behavior. Whether ZIv PV+ neurons are involved specifically in tactile-auditory integration or multisensory integration in general remains to be further investigated.

Different ZI sectors are dominated by distinct cell types (*Mitrofanis, 2005*; *Ricardo, 1981*). It has been suggested that different ZI cell types or sectors may exhibit different connectivity patterns (*Mitrofanis, 2005*), contributing to ZI's multiplex roles in various physiological functions. For example, it has been shown that activation of GABAergic neurons in the rostral sector of ZI (ZIr), which are likely PV-negative, can induce binge-like eating via its projections to the periventricular nucleus of thalamus (*Zhang and van den Pol, 2017*), while Lhx6-expressing neurons in ZIv, which are also PV-negative, can regulate sleep through their projections to hypothalamic areas (*Liu et al., 2017*). Different sectors or cell types may also play different roles in defensive behavior. Indeed, we have previously shown that activation of ZIr GABAergic neurons reduces noise-induced flight via their projections to the periaqueductal gray (PAG) (*Chou et al., 2018*). This effect is opposite to that of activating ZIv PV+ neurons, which have few projections to PAG (*Figure 4—figure supplement 1*). More recently, using conditioned freezing response as a model, a study of ZIv PV+ neurons has shown that both silencing the PV+ neuron output and silencing the amygdala inhibitory input to the PV+ neurons disrupt fear memory acquisition as well as recall of remote fear memory (*Zhou et al., 2018*). In the current study, the behavior we examined is an innate defensive behavior. Therefore, ZIv PV+ neurons can play a role in regulating both innate and learned defensive behaviors, which are generated under different contexts and likely engage different upstream pathways. It would be

interesting to investigate in the future how ZI, through interactions among its different cell-types/subdivisions, regulates behaviors in complex sensory and behavioral environments.

We have identified POm as a target of ZIv PV+ neurons that is mainly responsible for the tactile enhancement of flight behavior. Silencing of the projection from ZIv PV+ neurons to POm prevents the faciliatory effect of tactile stimulation. PV+ neurons in ZI are GABAergic (*Barthó et al., 2002*) and provide inhibition to their target neurons. To achieve the effect of enhancing the behavioral output, disinhibitory circuits may be involved. POm is known to project to the striatum to modulate locomotion (*Ohno et al., 2012*; *Smith et al., 2012*). The inhibitory nature of striatal neurons makes them a good candidate for engaging disinhibition of distant output responses (*Grillner et al., 2005*; *Kreitzer and Malenka, 2008*). Furthermore, we have shown previously that the noise-induced flight behavior depends on a pathway from the auditory cortex (AC) to the cortex of inferior colliculus (ICx) and then to PAG (*Xiong et al., 2015*). It is possible that the ZIv-POm pathway directly or indirectly connect to midbrain areas downstream of the AC-ICx-PAG pathway (*Marchand and Hagino, 1983*; *Roseberry et al., 2016*). As such, somatosensory information carried by the ZIv-POm pathway can modulate the auditory-induced behavior mediated by the AC-ICx-PAG pathway. It would be interesting to investigate in the future whether and how the POm-striatal circuit is involved in this modulation.

Overall, ZI has complex input and output connectivity patterns (*Chou et al., 2018*; *Nicolelis et al., 1992*; *Roger and Cadusseau, 1985*; *Shammah-Lagnado et al., 1985*; *Zhou et al., 2018*). Through convergent and divergent connectivity with various brain areas, ZI may be able to carry out multiple physiological and behavioral functions synergistically.

# Materials and methods

**Key resources table**

| Reagent type (species) or resource | Designation | Source or reference | Identifiers | Additional information |
|---|---|---|---|---|
| Strain (mouse) | *Pvalb-ires-Cre* | Jachson Laboratory | Stock No. 008069 | |
| Strain (mouse) | Ai14 | Jachson Laboratory | Stock No. 007914 | |
| Strain (mouse) | C57BL/6J | Jachson Laboratory | Stock No. 000664 | |
| Recombinant DNA reagent | AAV2/1-CamKII-hChR2-eYFP-WPRE-hGh | UPenn Vector Core | | |
| Recombinant DNA reagent | AAV1-CAG-FLEX-eGFP-WPRE-bGH | UPenn Vector Core | | |
| Recombinant DNA reagent | AAV2/1-pEF1α-DIO-hChR2-eYFP | UPenn Vector Core | | |
| Recombinant DNA reagent | AAV1-CAG-FLEX-ArchT-GFP | UNC vector Core | | |
| Recombinant DNA reagent | pAAV-hSyn-hM4D(Gi)-mCherry | Addgene | Plasmid #50475 | |
| Other (stains) | NeuroTrace 640/660 Deep-Red Fluorescent Nissl Stain | Thermo Fisher | N21483 | IHC 1:500 |
| Chemical compound, drug | Muscimol | Thermo Fisher | M23400 | |

*Continued on next page*

*Continued*

| Reagent type (species) or resource | Designation | Source or reference | Identifiers | Additional information |
|---|---|---|---|---|
| Chemical compound, drug | Tetrodotoxin | Torcris | Cat. No. 1078 | 1 µM |
| Chemical compound, drug | 4-Aminopyridine (4-AP) | Torcris | Cat. No. 0940 | 1 mM |
| Chemical compound, drug | cyanquixaline (CNQX) | Sigma-Aldrich | C239 | 20 µM |
| Chemical compound, drug | clozapine-N-oxide (CNO) | Torcris | Cat. No. 4936 | 1 mg/kg IP; 3 µM local infusion; 5 µM in slice recording |
| Software | Offline Sorter | Plexon | version 4 | |
| Software | MATLAB | Mathworks | version R2017a | |

All experimental procedures used in this study were approved by the Animal Care and Use Committee at the University of Southern California. Male and female wild-type (C57BL/6) and transgenic (PV-*ires*-Cre; Ai14-tdTomato) mice aged 8–16 weeks were obtained from the Jackson Laboratory. Mice were housed on 12 hr light/dark cycle, with food and water provided ad libitum.

## Viral and reagent injections

Viral injections were carried out as we previously described (*Ibrahim et al., 2016*; *Zingg et al., 2017*). Stereotaxic coordinates were based on the Allen Reference Atlas (www.brain-map.org). Mice were anesthetized using 1.5% isoflurane throughout the surgery procedure. A small incision was made on the skin after shaving to expose the skull. A 0.2 mm craniotomy was made, and virus was delivered through a pulled glass micropipette with beveled tip (~15 µm diameter) by pressure injection. For anterograde tracing, AAV2/1-CamKII-hChR2-eYFP-WPRE-hGh (UPenn Vector Core, $1.6 \times 10^{13}$ GC/ml) was injected into the SSp barrel field (30 nl total volume; AP −1.1 mm, ML +3.5 mm, DV −0.6 mm) of PV-*ires*-Cre::Ai14. AAV1-CAG-FLEX-eGFP-WPRE-bGH (UPenn Vector Core, $1.7 \times 10^{13}$ GC/ml) was injected into the ZI (30 nl total volume; AP −2.1 mm, ML +1.5 mm, DV −4.3 mm) of PV-*ires*-Cre mice. Animals were euthanized 3–4 weeks following the injection for examination.

For activity manipulations, AAV2/1-pEF1α-DIO-hChR2-eYFP (UPenn Vector Core, $1.6 \times 10^{13}$ GC/ml), AAV1-CAG-FLEX-ArchT-GFP (UNC Vector Core, $1.6 \times 10^{13}$ GC/ml), and pAAV-hSyn-hM4D(Gi)-mCherry (Addgene, $3 \times 10^{12}$ VC/ml) was injected bilaterally into ZI (100 nl for each site; AP −2.1 mm, ML +1.5 mm, DV −4.3 mm) of PV-*ires*-Cre mice. AAV1-CamKII-hChR2(E123A)-eYFP-WPRE-hGh (UPenn Vector Core, $1.6 \times 10^{13}$ GC/ml) was injected into SSp (30 nl total volume; AP +1.1 mm, ML −3.5 mm, DV −0.6 mm) of wild-type C57BL/6 mice. Viruses were expressed for at least three weeks. For silencing studies, muscimol (M23400; ThermoFisher) was injected unilaterally into SSp (100 nl total volume; AP +1.1 mm, ML −3.5 mm, DV −0.6 mm) or bilaterally into ZI (100 nl total volume; AP −2.1 mm, ML +1.5 mm, DV −4.3 mm) of wild-type mice.

## Histology, imaging and quantification

Animals were deeply anesthetized and transcardially perfused with phosphate buffered saline (PBS) followed by 4% paraformaldehyde. Brains were post-fixed at 4 ˚C overnight in 4% paraformaldehyde and then sliced into 150 µm sections using a vibratome (Leica, VT1000s). To reveal the cytoarchitectural information, brain slices were first rinsed three times with PBS for 10 min, and then incubated in PBS containing Nissl (Neurotrace 620, ThermoFisher, N21483) and 0.1% Triton-X100 (Sigma-Aldrich) for 2 hr. All images were acquired using a confocal microscope (Olympus FluoView FV1000). To quantify the relative strength of axonal projections of ZIv PV+ neurons in downstream structures,

serial sections across the whole brain were collected. Regions of interest were imaged at 10X magnification across the depth of the tissue (15 µm z-stack interval). For each brain, images were taken using identical laser power, gain and offset values. Fluorescence quantifications were performed by converting the images into monochromatic so that each pixel had a grayscale ranging from 0 to 255. Intensity value of the region of interest (200 × 200 pixel) was normalized to the baseline value. For each region of interest, three or more sections were imaged and averaged. The fluorescence density for each target structure was normalized for each animal and averaged across the animal group.

## Optogenetic preparation and stimulation

One week before the behavioral tests, animals were prepared as previously described (*Xiong et al., 2015*). Briefly, to optogenetically manipulate ZI neuron cell bodies, or ZI-POm, ZI-SC or SSp-ZI axon terminals, mice were implanted with fiber optic cannulas (200 µm ID, Thorlabs) two weeks after injecting ChR2 or ArchT virus (*Boyden et al., 2005*; *Chow et al., 2010*). The animal was anesthetized and mounted on a stereotaxic apparatus (Stoelting co.). Small holes (500 µm diameter) were drilled at a 20-degree angle relative to the vertical plane above ZI (AP −2.1 mm, ML ±1.5 mm, DV −4.3 mm), POm (AP −2.0 mm, ML ±1.5 mm, DV −3.0 mm) or SC (AP −4.0 mm, ML ±1.5 mm, DV −2.0 mm). The cannulas were lowered to the desired depth and fixed in place using dental cement. In the meantime, a screw for head fixation was mounted on the top of the skull with dental cement. Light from a blue LED source (470 nm, 10 mW, Thorlabs) was delivered at a rate of 20 Hz (20 ms pulse duration) via the implanted-cannulas using a bifurcated patch cord (Ø200 µm, 0.22 NA SMA 905, Thorlabs) for ChR2 or GFP control animals. The plastic sleeve (Thorlabs) securing the patch cord and cannula was wrapped with black tape to prevent light leakage. Light from a green LED source (530 nm, 10 mW, Thorlabs) for ArchT animals was delivered continuously for 5 s. Animals were allowed to recover for one week before behavioral tests. During the recovery period, they were habituated to the head fixation on the running plate. The head screw was tightly fit into a metal post while the animal could run freely on a flat rotating plate. Following testing sessions, animals were euthanized, and the brain was imaged to verify the locations of viral expression and implanted optic fibers. Mice with mistargeted viral injections or misplaced fibers were excluded from data analysis.

## Behavioral tests
### Head-fixed Flight Response

The test was conducted in a sound-attenuation booth (Gretch-Ken Industries, Inc). Sound stimulation, LED stimulation and data acquisition software were custom developed in LabVIEW (National Instruments). Each mouse was tested for one session per day which lasted no longer than two hours. During the behavioral session, the animal was head-fixed, and the speed of the running plate was detected with an optical shaft encoder (US Digital) and recorded in real time (*Xiong et al., 2015*; *Zhang et al., 2018*; *Zhou et al., 2014*). A 2 s or 5 s noise sound at 80 dB SPL (Scan-speaker D2905) was presented to trigger flight response as previously described. The stimulus was repeated for about 20 trials per session at an irregular interval ranging from 120 to 180 s. Little adaptation was observed (*Xiong et al., 2015*). Whisker stimulation (for 2 s) was delivered through a cotton stick controlled by a piezo actuator (Thorlabs). The stimulation frequency was 5 Hz and the vibration range was 4 mm. For optogenetic experiments, the blue or green LED stimulation (lasting for the entire 5 s duration of noise presentation) was randomly co-applied in half of the trials. LED-On and LED-Off trials were interleaved. The exact sequence, 'On-Off-On-Off...' or 'Off-On-Off-On...", was randomized for animals in the same group, or between different test sessions. Whisker stimulation was applied on the same side of auditory stimulation during the 2 s noise presentation without or with muscimol infusions into the contralateral SSp or bilateral ZI. For DREADDi experiments, animals infected with AAV-hM4Di(Gi)-mCherry (*Zhu and Roth, 2014*) received either an intraperitoneal (IP) injection of clozapine-N-oxide (CNO) (1 mg/kg), or a local infusion of CNO (3 µM, 100 nl) (*Zhu et al., 2016*) or saline (100 nl) through implanted cannulas into the POm. For the LED-only or whisker stimulation only control experiments, LED or whisker stimulation was given in the same way but without noise stimulation. Each animal was tested for consecutive 2 days and data were averaged across days for each animal.

## Two-Chamber Flight

C57LB/6 mice were placed inside a two-chamber test box (25 cm ×40 cm × 25 cm for each chamber). The two chambers were connected by a 50 cm long and 4 cm wide channel. Animals were allowed to habituate in the arena for 10 min. 10 s 80 dB SPL noise was delivered in one of the chambers. Animals flee to the other chamber by crossing the channel, which was video recorded. Each animal was tested for two consecutive days (two trials per day). On day two, 5 hr before the testing session, all whiskers were trimmed under anesthesia using 1.5% isoflurane throughout the procedure.

## Slice preparation and recording

To confirm the connectivity between SSp axons and ZI PV+ neurons. PV-*ires*-Cre::Ai14 mice injected with AAV2/1-pEF1α-DIO-hChR2-eYFP in SSp were used for slice recording. Three weeks following the injections, animals were decapitated following urethane anesthesia and the brain was rapidly removed and immersed in an ice-cold dissection buffer (composition: 60 mM NaCl, 3 mM KCl, 1.25 mM NaH$_2$PO$_4$, 25 mM NaHCO$_3$, 115 mM sucrose, 10 mM glucose, 7 mM MgCl$_2$, 0.5 mM CaCl$_2$; saturated with 95% O$_2$ and 5% CO$_2$; pH = 7.4). Coronal slices at 350 μm thickness were sectioned by a vibrating microtome (Leica VT1000s), and recovered for 30 min in a submersion chamber filled with warmed (35°C) ACSF (composition:119 mM NaCl, 26.2 mM NaHCO$_3$, 11 mM glucose, 2.5 mM KCl, 2 mM CaCl$_2$, 2 mM MgCl$_2$, and 1.2 NaH$_2$PO$_4$, 2 mM Sodium Pyruvate, 0.5 mM VC). ZIv neurons surrounded by EYFP$^+$ fibers were visualized under a fluorescence microscope (Olympus BX51 WI). Patch pipettes (~4–5 MΩ resistance) filled with a cesium-based internal solution (composition: 125 mM cesium gluconate, 5 mM TEA-Cl, 2 mM NaCl, 2 mM CsCl, 10 mM HEPES, 10 mM EGTA, 4 mM ATP, 0.3 mM GTP, and 10 mM phosphocreatine; pH = 7.25; 290 mOsm) were used for whole-cell recordings. Signals were recorded with an Axopatch 700B amplifier (Molecular Devices) under voltage clamp mode at a holding voltage of –70 mV for excitatory currents, filtered at 2 kHz and sampled at 10 kHz (*Ji et al., 2016*). Tetrodotoxin (TTX, 1 μM) and 4-aminopyridine (4-AP, 1 mM) were added to the external solution for recording monosynaptic responses only (*Petreanu et al., 2009*) to blue light stimulation (5 ms pulse, 3 mW power, 10–30 trials). CNQX (20 μM, Sigma-Aldrich) was added to the external solution to block glutamatergic currents.

For testing the efficacies of ChR2, ArchT and DREADDi, brain slices were prepared similarly, and whole-cell current-clamp recordings were made in neurons expressing ChR2, ArchT or DREADDi. A train of blue light pulses (20 Hz, 5 ms pulse duration) was applied to measure spike responses of ChR2- expressing neurons. Green light stimulation (500 ms duration) was applied to measure hyperpolarizations in ArchT-expressing neurons. For neurons expressing DREADDi receptors, a series of 500 ms current injections with amplitude ranging from 0 to 200 pA in 20 pA steps were applied before and after perfusion of CNO (5 μM) and after washing out CNO.

## Optrode recording and spike sorting

The mouse was anesthetized with isoflurane (1.5%–2% by volume), and a head post for fixation was mounted on top of the skull with dental cement and a craniotomy was performed over ZI (AP −2.0 ~ −2.2 mm, ML +1.4 ~ +1.6 mm) three days before the recording. Silicone adhesive (Kwik-Cast Sealant, WPI Inc) was applied to cover the craniotomy window until the recording experiment. Recording was carried out with an optrode (A1 × 16-Poly2-5mm-50 s-177-OA16LP, 16 contacts separated by 50 μm, the distance between the tip of the optic fiber and the probes is 200 μm, NA 0.22, Neuronexus Technologies) connected to a laser source (473 nm) with an optic fiber. The optrode was lowered into the ZIv region, and data were acquired with the Plexon recording system. The PV + neurons were optogenetically tagged by injecting floxed AAV-ChR2 in PV-Cre animals. To identify PV+ neurons, 20 Hz (20 ms pulse duration, 500 ms total duration) laser pulse trains were delivered intermittently. Signals were recorded and filtered through a bandpass filter (0.3–3 kHz). The nearby four channels of the probe were grouped as tetrodes, and semiautomatic spike sorting was performed by using Offline Sorter (Plexon). Semiautomated clustering was carried out on the basis of the first three principal components of the spike waveform on each tetrode channel using a T-Dist E-M scan algorithm (scan over a range of 10–30 degree of freedom) and then evaluated with sort quality metrics. Clusters with isolation distance <20 and L-Ratio > 0.1 were discarded (*Zhang et al., 2018*). Spike clusters were classified as single units only if the waveform SNR (Signal Noise Ratio)

exceeded 4 (12 dB) and the inter-spike intervals exceeded 1.2 ms for >99.5% of the spikes. To assess whether these units were driven directly by ChR2 or indirectly by synaptic connections, we analyzed the onset latency relative to each light stimulation. Only spikes with latency <3 ms were considered as being directly stimulated in this study. The whisker, noise or LED stimulation was given in a pseudorandom order for 7 to 12 trials. The evoked firing rate was calculated within the stimulation time window, subtracting the spontaneous firing rate.

## Data processing

For the head-fixed running test, running speed was recorded at 10 Hz sampling rate. The code for analyzing running speed is available at https://github.com/xiaolinchou/flight-speed-calculation, copy archived at https://github.com/elifesciences-publications/flight-speed-calculation). For each animal, trials were excluded if the peak noise-induced speed did not exceed the baseline speed by three standard deviations. Peak speed was determined as the maximum running speed after averaging all running trials. Total travel distance was calculated as the integral of running speed within a 5 s window after the onset of noise. Significance was tested between two conditions for all running trials, considering the trial-by-trial variation. For the two-chamber flight test, flight speed was calculated as the length of the channel divided by the total time animal spent in it.

## Statistics

Shapiro–Wilk test was first applied to examine whether samples had a normal distribution. In the case of a normal distribution, two-tailed t-test or one-way ANOVA test was applied. Statistical analysis was conducted using SPSS (IBM) and Excel (Microsoft).

# Acknowledgements

This work was supported by grants from the US National Institutes of Health (EY019049 and EY022478 to HWT; R01DC008983 to LIZ). HWZ was also supported by the Kirchgessner Foundation.

# Additional information

## Funding

| Funder | Grant reference number | Author |
| --- | --- | --- |
| National Institutes of Health | DC008983 | Li I Zhang |
| National Institutes of Health | EY019049 | Huizhong W Tao |
| Karl Kirchgessner Foundation | | Huizhong W Tao |
| National Institutes of Health | EY022478 | Huizhong W Tao |

The funders had no role in study design, data collection and interpretation, or the decision to submit the work for publication.

## Author contributions

Xiyue Wang, Xiaolin Chou, Conceptualization, Data curation, Formal analysis, Investigation; Bo Peng, Li Shen, Junxiang J Huang, Data curation; Li I Zhang, Huizhong W Tao, Conceptualization, Funding acquisition, Writing—review and editing

## Author ORCIDs

Xiyue Wang (iD) https://orcid.org/0000-0002-5805-0778
Huizhong W Tao (iD) https://orcid.org/0000-0002-3660-0513

## Ethics

Animal experimentation: All experimental procedures used in this study were approved by the Animal Care and Use Committee at the University of Southern California under the protocol 20719.

Decision letter and Author response
Decision letter https://doi.org/10.7554/eLife.42728.021
Author response https://doi.org/10.7554/eLife.42728.022

## Additional files

### Supplementary files

• Supplementary file 1. Analysis of temporal profiles of speed traces in different sets of experiments. Data are presented as mean ± SD. Two-sided paired t-test were performed to compared values between control and manipulation conditions. The type of experiment is shown by the corresponding figure number in main figures.
DOI: https://doi.org/10.7554/eLife.42728.018

• Transparent reporting form DOI: https://doi.org/10.7554/eLife.42728.019

### Data availability

All data generated or analysed during this study are included in the manuscript and supporting files. The data for each figure have been provided as source data files and the code used for data analysis can be found at https://github.com/xiaolinchou/flight-speed-calculation (copy archived at https://github.com/elifesciences-publications/flight-speed-calculation).

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
