## [Decision Letter]

Thank you for sending your article entitled "A Cross-modality Enhancement of Defensive Flight via Parvalbumin Neurons in Zona Incerta" for peer review at *eLife*. Your article has been evaluated by three peer reviewers, and the evaluation has been overseen by Andrew King as the Senior and Reviewing Editor.

The reviewers agreed that this is a well-conducted and interesting study, but have raised a number of important concerns. Given the list of essential revisions, including new experiments, the editors and reviewers invite you to respond within the next two weeks with an action plan and timetable for the completion of the additional work. We plan to share your responses with the reviewers and then issue a binding recommendation.

It was agreed that you have not included an adequate validation of the optogenetic manipulations, particularly as the effects reported are relatively modest, and that it is important to show in vivo how tactile and auditory inputs interact to determine the responses of ZI neurons. The reviewers also raised queries about the downstream projections from these neurons and how the results reported fit with previous studies. Finally, doubts were raised about the ecological relevance of tactile facilitation of auditory behavior.

*Reviewer #1:*

In this study, the authors tried to explore the role of ZI PV+ neurons in the cross-modal enhancement of defensive flight behavior. Although this appears to be an interesting study, the data at this point are rather preliminary. Several key experiments are lacking before the manuscript become convincing. This reviewer believes the authors should perform the following suggested experiments.

1) The authors have to record the activity of ZI PV+ neurons in vivo, either by fiber photometry or by single-unit recording. This type of experiment is regular and must-have for studies of neural circuits. Do ZI PV+ neurons respond to whisker deflections? Are they activated by sound that trigger flight responses? How do they integrate the sensory stimuli with different modalities?

2) The authors should expand their optogenetic experiments. For example, in Figure 2C-E, they should apply photostimulation with different light power and frequency. The phenotypes here appear to be extremely small. Would photostimulation with higher power or frequency be able to induce stronger differences between LED ON and OFF?

3) The authors need to provide a more detailed description of the projection targets of ZI PV+ neurons. The ZI project to multiple brain areas that control binge-like eating (Zhang and van den Pol, 2017) and sleep regulation (Liu et al., 2017). Considering these recent reports, this reviewer is not convinced that the ZI PV+ neurons only have two downstream targets that deserve scrutinizing.

4) As to the POm and SC as the postsynaptic targets of the ZI PV+ neurons, the authors need to answer two additional questions. What percentage of ZI PV+ neurons send collaterals to the POm and SC? If the POm-projecting ZI PV+ neurons are specifically silenced, what will happen?

5) The profiles of locomotion speed appear to be very zig-zag. This reviewer strongly suggest that the authors provide more analyses to compare the temporal properties of mouse escape behaviors in control and test groups.

6) As an essential part of data, the author need to present slice physiological data or in vivo recording data to functionally validate the ChR2, ArchT and hM4D to manipulate neuronal activity.

7) Again, as an essential part of data, the author need to provide how they validate the specificity of genetic manipulations of ZI PV+ neurons. For example, an immunohistochemical labeling of PV antibody and the fluorescent ChR2, ArchT and hM4D (Gi) are required.

8) The authors need to discuss whether the cross-modal property of ZI PV+ neurons is only specific for somatosensory-auditory integration. Alternatively, are they a bona fide node for any multisensory integration?

9) In this study, the author stated that "tactile stimulation was […] controlled by a piezo actuator (see Materials and methods)". However, the description of their piezo actuator is missing in Materials and methods. The authors need to add this part in Materials and methods and provide a supplementary figure to illustrate how they provide whisker stimulation. The intensity and frequency of whisker stimulation sometimes are critical factors to determine the valence of somatosensory stimuli. Therefore, the authors should be very careful to optimize their stimulation mode.

*Reviewer #2:*

The manuscript describes the results of an interesting set of experiments designed to examine somatosensory-mediated facilitation of auditory induced avoidance behavior in the mouse, and the neural circuits supporting this facilitated behavior. The authors show that vibrissal stimulation increases the speed of escape behaviors elicited by a loud sound. They go on to further show through a series of elegant experiments using anatomical, pharmacological, optogenetic and chemogenetic methods that the behavior is mediated by projections from somatosensory cortex to a subregion of zone incerta, and that parvalbumin expressing neurons in the ZI and that project to the medial posterior nucleus of the thalamus are the critical circuit elements. Overall the paper is reasonably well-written (it could use an editorial pass to improve some of the language) and experiments appear to be well-designed and well-conducted. My major questions regarding the paper have to do with the ecological validity of the behavior being examined, and how this behavior was tested.

First although I completely understand the concept of multisensory facilitation of behavior, I'm struggling to see the adaptive value of what is being measured. Under what circumstances would a loud sound and light touch to the vibrissae happen that would necessitate escape? Second, without quantifying the direction of the running, how do we know whether this is approach of avoidance? Third, why is running speed the measure of interest? From an adaptive perspective, the critical variable here would seem to be latency of response (i.e., how quickly is the escape triggered). Finally, and related to the above, without any meaningful spatial information in the stimuli (the sound appears to be very generalized – and paired with a touch to a left whisker), how can this be called escape? This is much more reminiscent of an acoustic startle paradigm, where multisensory facilitation has been well established.

As a final general comment, although the results of the ontogenetic experiments all go in the predicted direction, I am not sure of how robust the results are or of the statistical approaches that have been used to detail the differences.

The results are of interest – I am just trying to place them in a comprehensible behavioral context and am struggling without further elaboration and discussion.

*Reviewer #3:*

In this paper, Wang et al. shows that somatosensory input enhances sound-induced flight behavior through PV+ neuron of ZIv. It is an interesting topic that flight behavior can be enhanced by multisensory input, but there are several key concerns.

1) The authors discussed that ZIv-POm can project to the striatum to influence locomotion directly. If so, they should have seen that the direct activation of this circuit enhance the locomotion. In Figure 4—figure supplement 1, authors did not see the effect (although the flight speed tended to increase overall). Then, how the POm projection could enhance the flight response? In their previous paper, it was shown that a circuit projecting from ACx to ICx to PAG drives noise-induced flight response directly. The POm projection should be integrated into this ACx downstream pathway to enhance the audition-induced flight response. Projection from POm must enhance the auditory ascending pathway or strengthen ACx-to-ICx circuit. Further experiments or explanations need to clarify this to understand how the flight behavior (output of PAG) is modulated by the 'cross-modal' interaction between audition and somatosensation.

2) The authors claim that the SSp axon terminal is well observed in ZIv and can be targeted easily for optogenetic manipulation. However, the optogenetic manipulation effect actually seems very weak. The authors described in the method that they measured the noise-induced locomotion in 20 trials across 2 sessions and averaged all. In the average trace, the degree of LED light-induced improvement effect looks smaller than the variation in the control conditions (e.g. the peak speeds at LED OFF in Figure 3C and F). The authors need to clarify the effect of optogenetic manipulation of ZIv-PV+ neurons by statistically analyzing the significance of the LED light-ON effect against trial-by-trial behavioral variation in a mouse. Were LED-ON and LED-OFF trials randomly presented throughout the experiment?

3) In addition, the authors need to prove the presence of excitatory synaptic inputs from the SSp to PV+ neurons of ZIv through slice patch clamp recordings or similar electrophysiology methods.

---

## [Author Response]

[Editors' note: the authors’ plan for revisions was approved and the authors made a formal revised submission.]

Reviewer #1:

*[…] 1) The authors have to record the activity of ZI PV+ neurons* in vivo*, either by fiber photometry or by single-unit recording. This type of experiment is regular and must-have for studies of neural circuits. Do ZI PV+ neurons respond to whisker deflections? Are they activated by sound that trigger flight responses? How do they integrate the sensory stimuli with different modalities?*

We have performed awake optrode recording to examine responses of ZI PV+ neurons to whisker and noise stimulation. We found that ZI PV+ neurons are responsive to both noise sound and whisker deflections. In addition, the noise responses of these neurons are enhanced by concurrent whisker deflections (Figure 3J-L). Therefore, cross-modality integration takes place in ZI PV+ neurons, in which somatosensory input has a faciliatory role.

2) The authors should expand their optogenetic experiments. For example, in Figure 2C-E, they should apply photostimulation with different light power and frequency. The phenotypes here appear to be extremely small. Would photostimulation with higher power or frequency be able to induce stronger differences between LED ON and OFF?

We agree with the reviewer that the effect is only moderate. For Figure 2C-E, in the revision, we have increased the animal number and the result now shows an increased significance level. There are two factors that may explain the relatively small effect. First, we have used white noise at a high intensity of 80dB SPL, which could trigger a high-magnitude (possibly close to saturation) flight response according to our previous study (Xiong et al., 2015). Second, the activation of terminals may not be highly efficient, and the axon projections may only be partially labeled.

In fact, we have used the maximum LED power available in our current system and a very high stimulation frequency (20 Hz) that ZI PV+ neurons can fire (see Figure 3L). To test whether a stronger modulatory effect can be obtained, we then applied a lower noise intensity (60dB SPL). We found a large increase (127% ± 121%, n = 4) in the probability of sound-evoked running in LED-On vs LED-Off trials, although when only running trials were analyzed the percentage increase of peak speed in LED-On trials was not significantly larger than in the 80-dB noise condition. We think this result does suggest a stronger faciliatory effect if a lower-intensity noise is used as the stimulus.

3) The authors need to provide a more detailed description of the projection targets of ZI PV+ neurons. The ZI project to multiple brain areas that control binge-like eating (Zhang and van den Pol, 2017) and sleep regulation (Liu et al., 2017). Considering these recent reports, this reviewer is not convinced that the ZI PV+ neurons only have two downstream targets that deserve scrutinizing.

In the revised manuscript, we have provided a complete summary of downstream targets of ZI PV+ neurons (Figure 4—figure supplement 1), which shows the relative fluorescence density of axon labeling in different targets. From this figure, it is clear that POm and SC are the two strongest targets of ZI PV+ neurons. We have performed additional experiments to show that locally silencing the ZI PV+ axons in POm blocks the facilitatory effect of whisker stimulation (Figure 4I-K). This further confirms the involvement of the ZIv-POm pathway. It is worth noting that Zhang and van den Pol, 2017 paper studied ZIr neurons and Liu et al., 2017 paper studied ZIv Lhx+ neurons which are PV negative. Together, these studies support the view that different cell types in ZI may play distinct functional roles in different types of behavior.

4) As to the POm and SC as the postsynaptic targets of the ZI PV+ neurons, the authors need to answer two additional questions. What percentage of ZI PV+ neurons send collaterals to the POm and SC? If the POm-projecting ZI PV+ neurons are specifically silenced, what will happen?

The reviewer’s concern over specificity is warranted. In the revision, we specifically silenced the axonal terminals from ZI PV+ neurons in POm by expressing DREADDi receptors in ZI PV+ neurons and injecting CNO locally in POm, following previous studies (Zhu et al., 2016; Zhang et al., 2018). This blocked the modulatory effect of whisker stimulation on the noise-induced escape behavior (Figure 4I-K), suggesting that the ZI PV-POm projection is primarily responsible for the tactile effect.

5) The profiles of locomotion speed appear to be very zig-zag. This reviewer strongly suggest that the authors provide more analyses to compare the temporal properties of mouse escape behaviors in control and test groups.

The reviewer has raised a good point. Following the reviewer’s suggestion, we have analyzed the onset latency of the running behavior (determined by the time point at which the speed exceeds the baseline value by 3 standard deviations) as well as the time to peak speed. Our results did not reveal any significant differences between control and manipulation conditions in any set of experiments (see Supplementary file 1).

6) As an essential part of data, the author need to present slice physiological data or in vivo recording data to functionally validate the ChR2, ArchT and hM4D to manipulate neuronal activity.

In the revision, we have provided electrophysiological validations of effects of activating ChR2, ArchT and hM4D on membrane potential responses, using slice whole-cell recording methods (Figure 3—figure supplement 1B-C, 3A-C).

7) Again, as an essential part of data, the author need to provide how they validate the specificity of genetic manipulations of ZI PV+ neurons. For example, an immunohistochemical labeling of PV antibody and the fluorescent ChR2, ArchT and hM4D (Gi) are required.

The PV-*ires*-Cre mouse we used for manipulating PV neuron activity has been widely used by many labs, with previous demonstrations of PV-specific expression (Madisen et al., 2010; Zhou et al., 2018). With injections of Cre-dependent viruses, the specificity of expression of opsins or DREADD receptors in PV+ neurons is not a concern. In the revision, we have provided representative images showing co-localization of ChR2-EYFP with tdTomato in PV-Cre crossed with Ai14 (Cre-dependent tdTomato reporter) mice (Figure 3—figure supplement 1A).

8) The authors need to discuss whether the cross-modal property of ZI PV+ neurons is only specific for somatosensory-auditory integration. Alternatively, are they a bona fide node for any multisensory integration?

In the current study, we only provide evidence that ZI PV+ neurons are involved in somatosensory-auditory interaction, an example of cross-modality interaction. Whether these neurons could be a bona fide node for any multisensory integration is unknown and requires more future investigations. We have clarified this point in the Discussion (third paragraph).

9) In this study, the author stated that "tactile stimulation was….controlled by a piezo actuator (see Materials and methods)". However, the description of their piezo actuator is missing in Materials and methods. The authors need to add this part in Materials and methods and provide a supplementary figure to illustrate how they provide whisker stimulation. The intensity and frequency of whisker stimulation sometimes are critical factors to determine the valence of somatosensory stimuli. Therefore, the authors should be very careful to optimize their stimulation mode.

In the revised manuscript, we have added information on the piezo as well as the frequency (5Hz) and vibration range (4 mm) of whisker deflections in Materials and methods (subsection “Head-fixed Flight Response”). We have also added a photograph of the whisker stimulation apparatus to show the detail (Figure 1—figure supplement 1A).

Reviewer #2:[…] First although I completely understand the concept of multisensory facilitation of behavior, I'm struggling to see the adaptive value of what is being measured. Under what circumstances would a loud sound and light touch to the vibrissae happen that would necessitate escape? Second, without quantifying the direction of the running, how do we know whether this is approach of avoidance? Third, why is running speed the measure of interest? From an adaptive perspective, the critical variable here would seem to be latency of response (i.e., how quickly is the escape triggered). Finally, and related to the above, without any meaningful spatial information in the stimuli (the sound appears to be very generalized – and paired with a touch to a left whisker), how can this be called escape? This is much more reminiscent of an acoustic startle paradigm, where multisensory facilitation has been well established.

We apologize for the insufficient background information provided in the previous manuscript. In our previous study (Xiong et al., 2015), we have demonstrated that noise sound can evoke an escape behavior in freely moving animals, manifested by their immediate running away from the chamber containing the sound source. This behavior can be much more easily quantified by using the head-fixed preparation (Xiong et al., 2015). When sound is applied on either side of the animal, it always exhibits forward running in the head-fixed condition. In the current study, the sound source is on the same side of whisker stimulation. This has now been clarified in Materials and methods (subsection “Head-fixed Flight Response”).

The reviewer has raised a good point about ecological validity. In general, whiskers are essential for rodents to gather information from surroundings. When escape behavior is concerned, loud noise plus specific tactile input may indicate the immediate proximity of a predator, and enhancement of flight at such moments will greatly increase survival chances of prey animals. In addition, tactile information during escape may help animals to quickly identify the most efficient escape route and thus facilitate the escape. We have added more discussion about these possibilities in the revised manuscript (Discussion, second paragraph). Furthermore, we have performed a new set of experiments in freely moving animals. We found that trimming of all whiskers significantly reduced the speed of the animal when it escaped from the sound source by crossing a narrow tunnel (Figure 1—figure supplement 2). This suggests that tactile input through whiskers can indeed enhance the efficiency of escape in a more natural condition.

We previously showed that the peak speed can be a measure of the escape behavior as it is positively correlated with the intensity of noise sound applied (Xiong et al., 2015). Nevertheless, the efficiency of escape is determined not only by the response latency and peak speed, but also by the total travel distance within a certain time window. In the revision, we have analyzed the temporal profile of the behavioral response, including onset latency and time to peak speed, but found no significant differences between control and manipulation conditions in any set of experiments (see Supplementary file 1). However, when the total travel distance (shown as “average speed” in the previous manuscript) or peak speed is concerned, there is a significant difference.

As a final general comment, although the results of the ontogenetic experiments all go in the predicted direction, I am not sure of how robust the results are or of the statistical approaches that have been used to detail the differences.The results are of interest – I am just trying to place them in a comprehensible behavioral context and am struggling without further elaboration and discussion.Reviewer #3:[…] 1) The authors discussed that ZIv-POm can project to the striatum to influence locomotion directly. If so, they should have seen that the direct activation of this circuit enhance the locomotion. In Figure 4—figure supplement 1, authors did not see the effect (although the flight speed tended to increase overall). Then, how the POm projection could enhance the flight response? In their previous paper, it was shown that a circuit projecting from ACx to ICx to PAG drives noise-induced flight response directly. The POm projection should be integrated into this ACx downstream pathway to enhance the audition-induced flight response. Projection from POm must enhance the auditory ascending pathway or strengthen ACx-to-ICx circuit. Further experiments or explanations need to clarify this to understand how the flight behavior (output of PAG) is modulated by the 'cross-modal' interaction between audition and somatosensation.

That activation of the ZIv-POm projection alone does not directly induce locomotion is possibly because the ZIv output is inhibitory and only plays a modulatory role.

The reviewer has raised a good question about where the facilitatory cross-modal interaction occurs. In the revision, we carried out additional in vivo electrophysiology experiments and found that ZI PV+ neurons respond to both noise sound and whisker deflections. In addition, the response to noise can be enhanced by concurrent whisker deflections (Figure 3J-L). Therefore, the multisensory integration takes place in ZI PV+ neurons at least. Furthermore, we speculate that the ZIv-POm pathway may directly or indirectly connect to midbrain areas downstream of the AC-ICx-PAG pathway, thus the somatosensory information carried by the ZIv-POm pathway can modulate the auditory-induced behavior mediated by the AC-ICx-PAG pathway. We have discussed this possibility in the revised Discussion (fifth paragraph).

In addition, we have specifically silenced ZI PV+ neuron projections to POm using chemogenetics. This blocked the faciliatory effect of whisker deflections (Figure 4I-K), further strengthening our conclusion that the ZI-POm pathway is involved.

2) The authors claim that the SSp axon terminal is well observed in ZIv and can be targeted easily for optogenetic manipulation. However, the optogenetic manipulation effect actually seems very weak. The authors described in the method that they measured the noise-induced locomotion in 20 trials across 2 sessions and averaged all. In the average trace, the degree of LED light-induced improvement effect looks smaller than the variation in the control conditions (e.g. the peak speeds at LED OFF in Figure 3C and F). The authors need to clarify the effect of optogenetic manipulation of ZIv-PV+ neurons by statistically analyzing the significance of the LED light-ON effect against trial-by-trial behavioral variation in a mouse. Were LED-ON and LED-OFF trials randomly presented throughout the experiment?

The apparent weak effect of stimulation of SSp axons was possibly due to only partial ChR2 labeling of SSp axons projecting to ZIv. In the revision, we have increased the animal number and the result now shows an increased significance level (see new Figure 2D).

The reviewer has raised a valid point about the relatively large variations in control running speed between individual animals. Following the reviewer’s suggestion, we have added comparisons of LED-On and LED-Off trials for each individual animal, considering trial-by-trial variations (Figure 1—figure supplement 1C-D, Figure 2—figure supplement 1A-B, Figure 3—figure supplement 1D-G, Figure 4—figure supplement 2A-B). From these plots, it is clear that the majority of animals tested showed a significant difference between control and manipulation conditions.

LED-On and LED-Off trials were interleaved. The exact sequence, “On-Off-On-Off…” or “Off-On-Off-On…”, was randomized for animals in the same group, or between different test sessions. This has been clarified in Materials and methods (subsection “Head-fixed Flight Response”).

3) In addition, the authors need to prove the presence of excitatory synaptic inputs from the SSp to PV+ neurons of ZIv through slice patch clamp recordings or similar electrophysiology methods.

In the revision, we have performed slice whole-cell recording in PV-Cre::Ai14 animals with AAV-ChR2 injected in SSp. We demonstrate that ZI PV+ neurons indeed receive direct excitatory input from SSp axons (new Figure 2I-K).